# Long-range allosteric regulation of the human 26S proteasome by 20S proteasome-targeting cancer drugs

David Haselbach[1,*], Jil Schrader[1,*], Felix Lambrecht[1], Fabian Henneberg[1], Ashwin Chari[1] & Holger Stark[1]

The proteasome holoenzyme is the major non-lysosomal protease; its proteolytic activity is essential for cellular homeostasis. Thus, it is an attractive target for the development of chemotherapeutics. While the structural basis of core particle (CP) inhibitors is largely understood, their structural impact on the proteasome holoenzyme remains entirely elusive. Here, we determined the structure of the 26S proteasome with and without the inhibitor Oprozomib. Drug binding modifies the energy landscape of conformational motion in the proteasome regulatory particle (RP). Structurally, the energy barrier created by Oprozomib triggers a long-range allosteric regulation, resulting in the stabilization of a non-productive state. Thereby, the chemical drug-binding signal is converted, propagated and amplified into structural changes over a distance of more than 150 Å from the proteolytic site to the ubiquitin receptor Rpn10. The direct visualization of changes in conformational dynamics upon drug binding allows new ways to screen and develop future allosteric proteasome inhibitors.

[1] Department for Structural Dynamics, Max-Planck Institute for Biophysical Chemistry, Am Fassberg 11, 37077 Göttingen, Germany. * These authors contributed equally to this work. Correspondence and requests for materials should be addressed to A.C. (email: ashwin.chari@mpibpc.mpg.de) or to H.S. (email: holger.stark@mpibpc.mpg.de).

The proteasome holoenzyme is composed of the catalytic core particle (CP, 750 kDa) and in addition either one or two molecules of the regulatory particle (RP, 900 kDa), to form the 26S (1.6 MDa) and 30S (2.5 MDa) proteasome holoenzyme[1], respectively. The CP consists of four co-axially stacked rings of seven distinct α and β subunits, whereas the RP consists of an AAA+ ATPase assembly (Rpt1–6) and 12 non-ATPase subunits (Rpn1–3 and Rpn5–13)[2] (Fig. 1a,b). Its main task is the degradation of polyubiquitinated substrates. Consequently, cellular homeostasis including diverse functions such as the control of the cell division cycle, transcription regulation, protein quality control, apoptosis and many more pathways, depends on its proteolytic activity[3]. For the mechanistic understanding of its cellular activities and its therapeutic targeting in disease, the elucidation of high-resolution structures of the proteasome holoenzyme in complex with drugs are therefore of paramount importance. In particular, the structural impact of 20S inhibitors on the proteasome holoenzyme remains entirely elusive.

Here we determined the high-resolution structures of the human 26S proteasome holoenzyme bound to the chemotherapeutic Oprozomib and its apo form using single particle cryo-electron microscopy (cryo-EM) and additionally determined the dynamic properties of the proteasome. We identify a clear restriction of the conformational landscape of the holoenzyme upon drug binding using a newly developed method to map the conformational and energy landscape of the 26S proteasome. From the energy landscape, we can infer that drug binding introduces an energy barrier minimizing the possibility of the RP to rotate on the CP. This rotation however is necessary for deubiquitination of the substrate[4] and its translocation into the proteolytic core of the proteasome. We thus find a long-range allosteric regulation that spans more than 150 Å from the location of drug binding towards the ubiquitin recognizing and regulatory regions in the 19S subunit.

## Results

### Structure determination of the inhibited 26S proteasome.
The treatment of the proteasome holoenzyme with 20S inhibitors leads to stabilization and suppresses disassembly[5]. To address the question how 20S inhibitors affect the proteasome holoenzyme structure, we initially considered the conformational motions of the RP, which have been described previously (Fig. 1g)[4,6,7]. In essence, two conformational states have been described: a non-rotated- and a rotated-state in which the non-ATPase segments of the RP are rotated by up to 25° around the long axis of the 20S CP. This motion is coupled to the ATPase part of the RP with its Rpt4/Rpt5 coiled coil contacting the ubiquitin receptor Rpn10 (Supplementary Movie 1). Treating the 26S proteasome with either the drug Oprozomib or the natural product Epoxomicin, which both belong to the epoxyketone class of 20S proteasome inhibitors, we found the proteasome holoenzyme to be stabilized in the non-rotated state (Fig. 1h). The treatment of the 26S proteasome with 20S proteasome inhibitors leads to a stabilization of polyubiquitinated substrates[8], which may remain bound to the proteasome holoenzyme. We therefore investigated if polyubiquitinated substrates are accumulated in our proteasome preparations upon inhibition by drugs using anti-ubiquitin western Blot analysis. As shown in Supplementary Fig. 1, we have found no profound accumulation of polyubiquitinated substrates in our inhibited 26S proteasome preparation over non-treated controls. This indicates no correlation between the accumulation of polyubiquitinated substrates and proteasome inhibition. As a consequence, the allosteric regulation of RP rotation described in this manuscript is

exclusively dependent on inhibitor binding (Supplementary Fig. 1). We then proceeded to reconstruct the three-dimensional (3D) structure of both the non-inhibited and Oprozomib-inhibited 26S proteasome at 4.8 Å/3.8 Å resolution, respectively. To achieve this, we utilized the identical image processing and classification protocol (Supplementary Fig. 2) for the reconstruction with and without Oprozomib. Notably, in both non-inhibited and inhibited structures the structure of the proteasome holoenzyme strongly resembles (real-space correlation > 0.9) the two other high-resolution EM structures reported recently[8,9]. While additional density is absent in the β5 active sites of the non-inhibited structure (Fig. 1f), a clear density for Oprozomib is visible in the β5 active sites of the inhibited structure (Fig. 1e, Supplementary Fig. 3).

### Model of the Oprozomib-bound human 26S proteasome.
The Oprozomib–26S proteasome structure (Supplementary Tables 1 and 2) exhibits well-defined densities for the entire proteasome holoenzyme (with the exception of Rpn1, Supplementary Fig. 4), showing numerous amino-acid side chains in most parts of the molecule and relatively small variations in local resolution (Fig. 1c). Specifically, regions encompassing both β subunit rings and the α subunit ring of the CP bound to the RP, as well as the ATPase of the RP are resolved at a resolution range of 3.5–4.5 Å. The local resolution of the structure decreases with increasing distance from the CP (Fig. 1c) to the upper regions of the proteasome lid structure, which appear to be the most mobile regions within the proteasome. Owing to the visible side-chain densities, we built an accurate atomic model of the holoenzyme with the unequivocal assignment of amino acid registry for regions with B factors smaller than 110 Å$^2$ (Fig. 1d, Supplementary Fig. 5). The B-factors of the model correlate well with the local resolution differences visible in the EM density map (Fig. 1c,d).

### Long-range allosteric effects of drug binding to the proteasome.
We utilized only a relatively small subset of particles from both non-inhibited and Oprozomib-inhibited datasets (4% and 12%, respectively) to obtain reconstructions at high resolution described above, which are nearly identical in conformation. Thus, the differences between proteasomes with and without drug are expected to be manifested in the particle images that did not contribute to the high-resolution structures and more likely affect the conformational space adopted by the proteasome. To harness this information, we studied the conformational variations in the RP in a quantitative manner by extensive 3D classification (Supplementary Fig. 6), calculated the corresponding energy landscape (Fig. 2) and analysed how drug binding modifies this energy landscape of the RP. Briefly, we focused on the conformational variability in the RP by aligning all 26S 3D structures with respect to their 20S part only and applied principal component analysis (PCA) to reveal the eigenvectors as major modes of motion of the RP (see Fig. 2, Methods, Supplementary Note 1 and Supplementary Movie 1). The known particle number for each conformation allows the transformation of a conformational landscape into an energy landscape that describes the complexity of RP motion in a comprehensive, quantifiable manner. It also enables the direct visualization of the changes in proteasome dynamics upon Oprozomib binding (Fig. 2b, Supplementary Movie 1). According to this analysis, the energy landscape of the non-inhibited proteasome is rather flat and allows the RP to sample a large number of conformations making use of thermal energy only. In contrast, drug binding considerably decreases the available conformational space and creates an energy barrier for the molecules making it less likely to reach a fully rotated state. This is

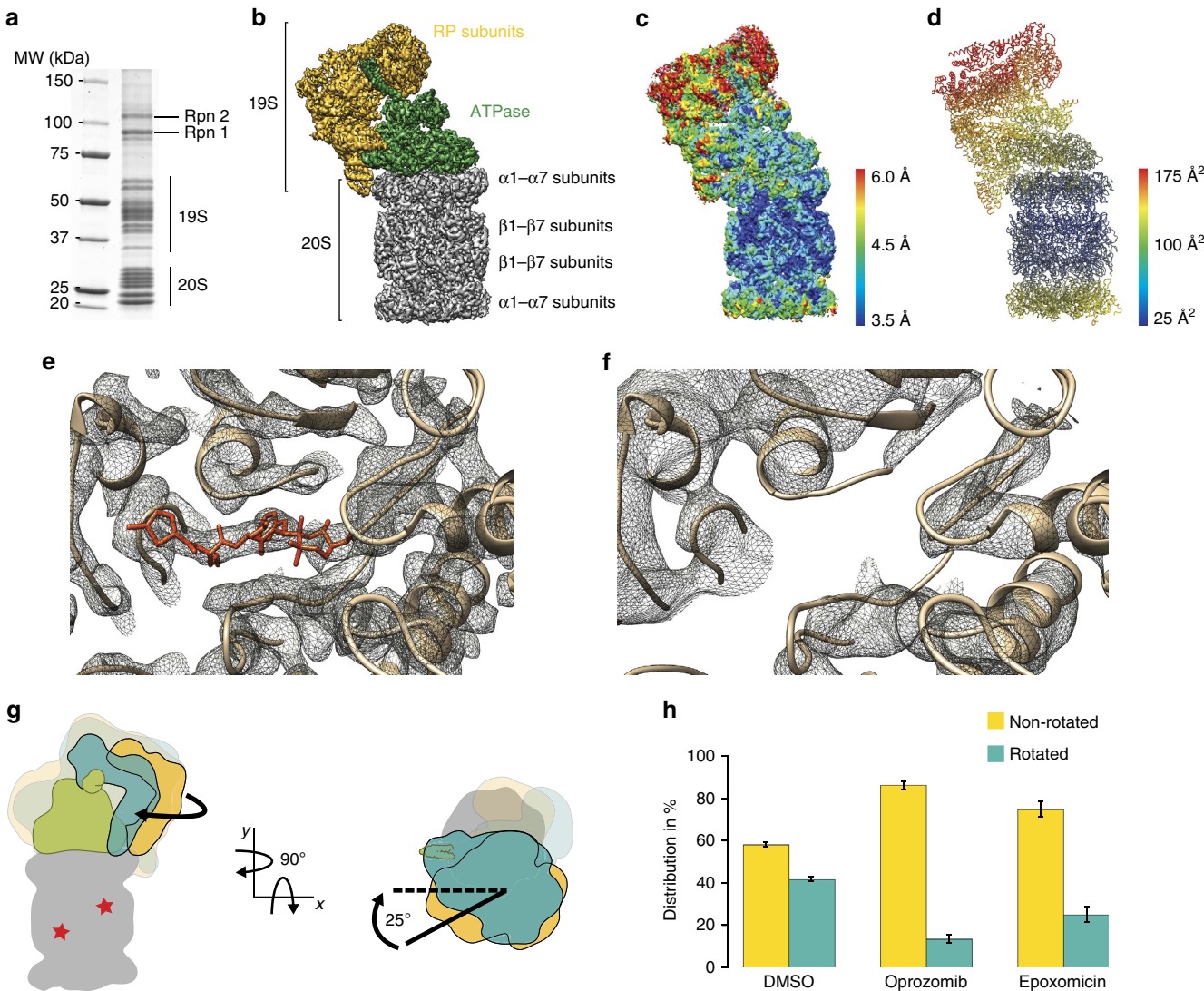

**Figure 1 | Effect of Oprozomib.** (**a**) SDS–PAGE of purified human proteasomes. (**b**) Surface view of the human Oprozomib-bound 26S proteasome cryo-EM density map at 3.8 Å resolution. The CP (20S) subcomplex is depicted in grey, the AAA+ ATPase subcomplex in green and the remaining RP (19S) components in yellow. (**c**) Local resolution map of the structure shown in **b** Each part of the density is coloured according to the local resolution as specified in the colour bar. The resolution ranges from 3.5 Å (blue) to 6 Å (red). (**d**) Atomic model of the complete 26S proteasome. The model is coloured according to the B-factor distribution. B factors range from 25 Å² (blue) to 175 Å² (red). (**e**) Close-up view of the Oprozomib binding site in the β5 subunit of the CP. The Oprozomib model is coloured in red the CP subunits are shown in brown. (**f**) Close-up view of the empty Oprozomib binding site in the β5 subunit of the CP. The Oprozomib model is coloured in red the CP subunits are shown in brown. (**g**) Schematic representation of the two major rotational modes of the RP reveals a rotation of the RP along the long axis of the 26S proteasome as indicated in a cartoon representation as a visual aid. (**h**) Histogram of the relative distribution of 26S proteasome particles found in either the rotated or the non-rotated state which can be modified by epoxyketone inhibitor binding. The control dataset (DMSO) reveals an almost balanced distribution with ∼41% of the particles in the rotated state. The number of particles in the rotated state is significantly reduced upon Oprozomib (∼13% rotated) or Epoxomicin (∼25% rotated) binding. Error bars displaying s.d. indicate a high reproducibility based on data from three independent proteasome preparations (n = 3).

manifested along several eigenvectors (data not shown) and indicates that although a subset (14%) of 26S proteasomes is observed in a rotated state upon Oprozomib inhibition, the maximal amplitude of rotation attained is only 20°. At this state of RP rotation, Rpn10 approaches the coiled coil of Rpt4/5, whereas non-inhibited 26S proteasomes rotate the RP up to 25° with high probability, where the Rpt4/5 coiled coil reaches the Rpn10/Rpn9 interface (Fig. 1g, Supplementary Movie 1).

Our data clearly suggests that in addition to a competitive, irreversible inhibition of the proteolytic activity[10], CP inhibitors also affect the conformational landscape of the proteasome holoenzyme by restriction of its capability to adopt the rotated state. The most surprising feature about the allosteric regulation

elicited by Oprozomib on the 26S proteasome is that the inhibition reaction is converted to a conformational signal that is relayed over a distance of over 150 Å.

## Discussion

In this paper, we have determined high-resolution structures of inhibited and non-inhibited endogenously purified 26S proteasomes. Inhibition of the proteasome allows a structure of higher resolution to be determined. Importantly, by employing a novel image analysis procedure, we can show that inhibitor binding causes a long-range allosteric regulation of the proteasome holoenzyme. Surprisingly, both species bind similar amounts of

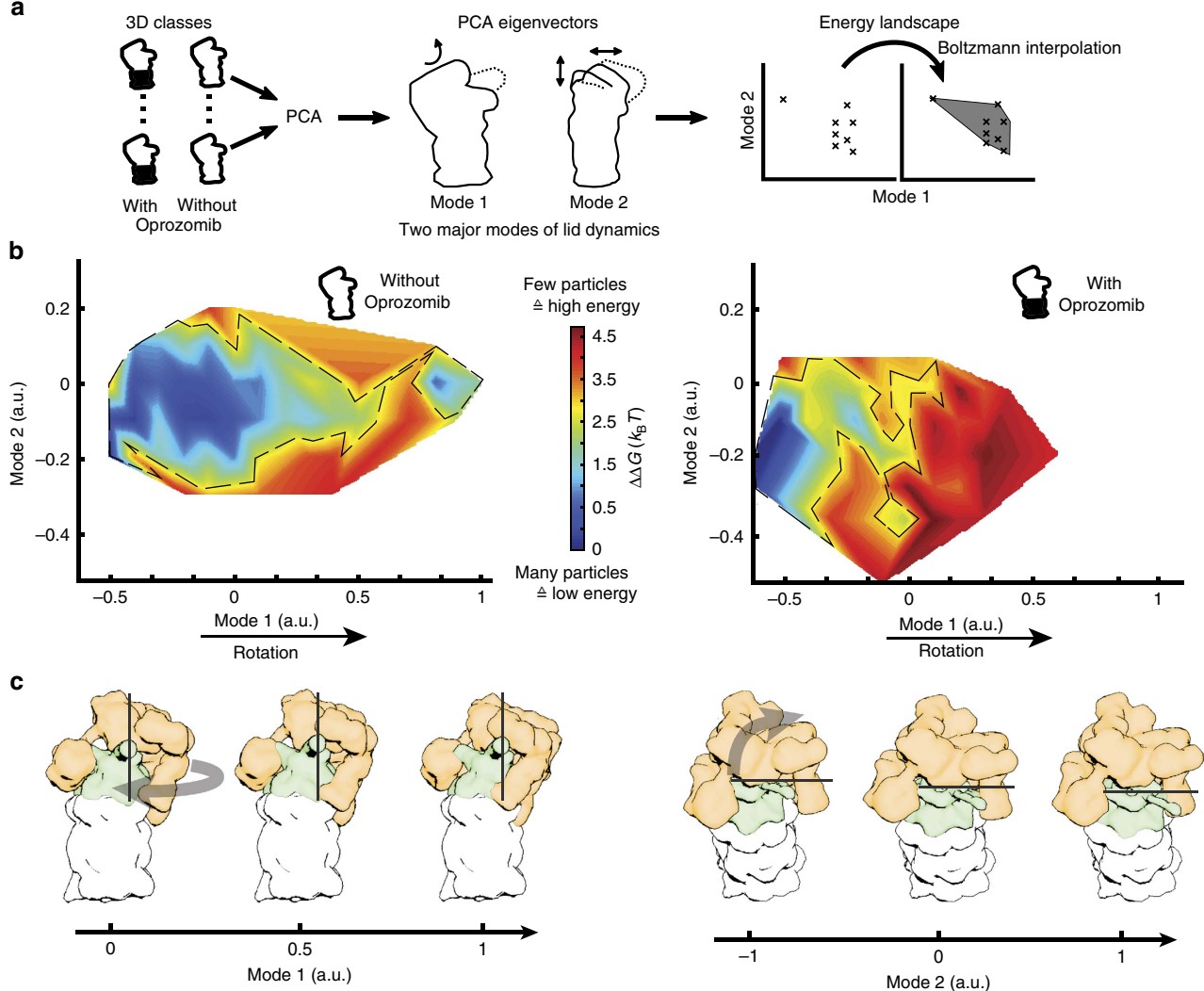

**Figure 2 | Modification of the energy landscape upon Oprozomib binding.** (**a**) Schematic representation of the strategy to obtain an energy landscape, which corresponds to the conformational states adopted by the proteasome holoenzyme: Principal component analysis of a large number ($n = 346$) of 3D structures yields eigenvectors corresponding to the major modes of conformational variability. The first mode corresponds to a lid rotation with respect to the 20S subunit and the second mode to a more complex rotational rearrangement of the lid (Supplementary Movies 1 and 2). The relative particle numbers, which can be obtained for the various conformational states, are used to calculate the energy landscape according to Boltzmann's law. (**b**) Energy landscapes with and without Oprozomib are depicted. Without Oprozomib, the energy landscape is rather broad and flat which allows proteasomes to sample a wide range of conformations without facing a significant energy barrier. In contrast, upon drug binding, well-separated minima can be observed next to a significant energy barrier (red) which restricts the conformational space that can be sampled by the proteasome. The 3.8 Å resolution structure was determined from particles belonging to this proteasome conformation in a local energy minimum (dark blue). (**c**) Graphical representation of the movement modes. The average structure has been segmented in its three subcomplexes and fitted as rigid bodies in the eigenmodes.

polyubiquitinated substrates as shown by western Blot analysis. We can therefore conclude that the observed long-range conformational changes of the proteasome upon drug binding are exclusively due to the binding of the inhibitor Oprozomib. This raises the interesting question about possible determinants that enable such long-range regulation. While an accurate description of this will require many more high-resolution structures, mechanistic biochemistry and molecular dynamics simulations, a detailed analysis of the Oprozomib-inhibited structure allows a first glimpse on the components that might be involved in signal relay and amplification (Supplementary Fig. 4). The chemical inhibition signal is located in the β-ring of the proteasome and needs to be relayed over several tiers (CP alpha ring, two ATPase rings) to the top of the lid structure of the RP. Two correlated criteria, such as decreased local resolutions in the EM reconstruction (Fig. 1c,d) and

correspondingly regions of higher B-factors in the atomic model (Supplementary Fig. 5) indicate conformational mobility, which we used to monitor the signal pathway from the inhibition site to the upper parts of the RP. Moving vertically in tiers from the site of inhibition (the 20S β5 active site), an asymmetric B-factor elevation on α-subunits 1, 2, 3 and 4 becomes evident (Fig. 3a). This asymmetric conformational mobility of the α-subunits is further relayed in two directions onto the RP: 1) A lateral transmission onto the adjacent RP subunits Rpn5 and 6 and 2) a vertical transmission to the next higher tier (Fig. 3b), the ATPase. At present, which pathway of transmission of conformational mobility (the lateral or vertical) is dominant remains unclear and speculative but both are most likely synergistic and mutually potentiate each other. Irrespective of the precise pathway of the propagation of conformational dynamics, the ATPase subunits (Rpt2, 3 and 6) adjacent to Rpn5 and 6 and directly above the

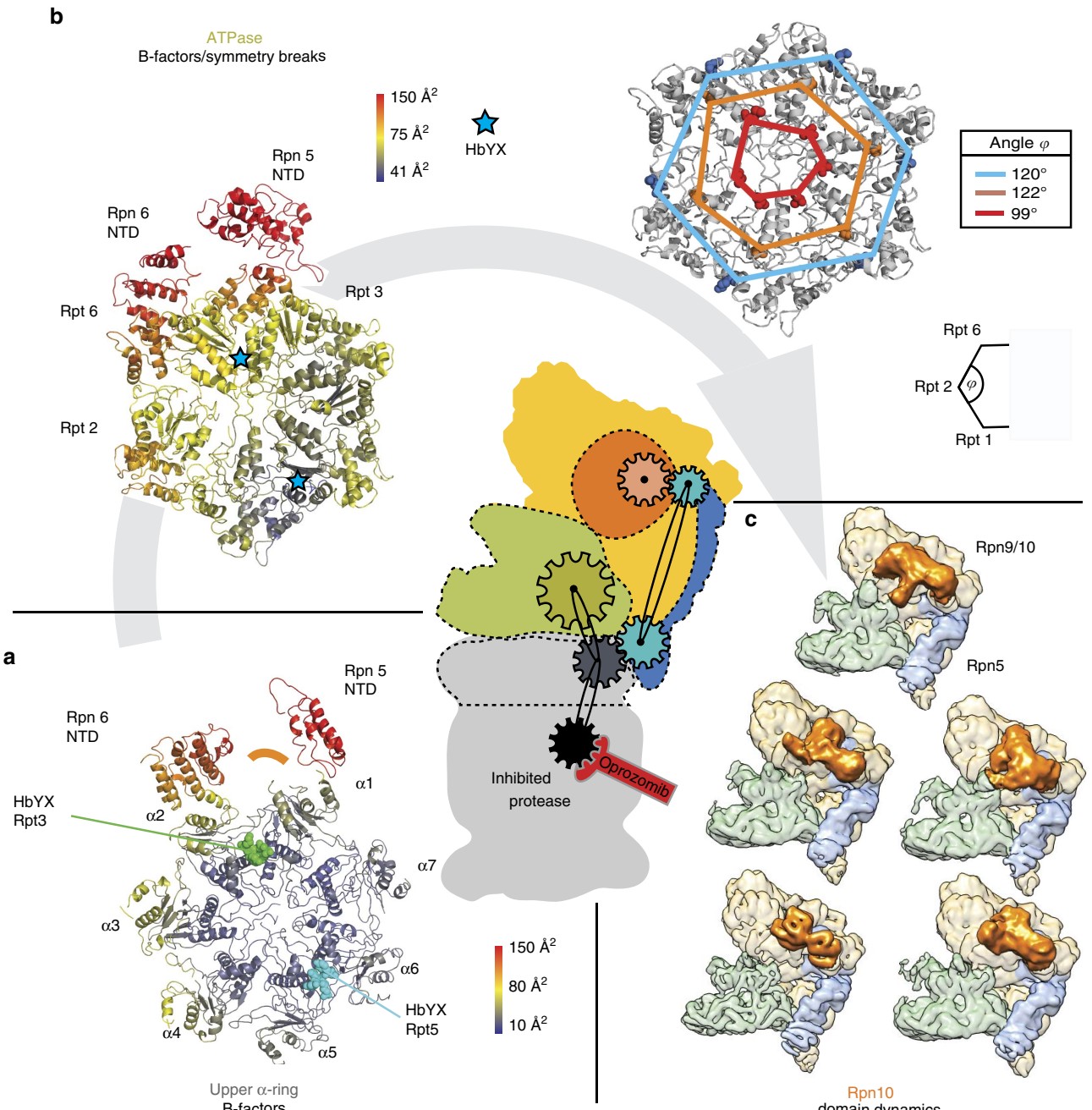

**Figure 3 | Representation of potential bi-directional signalling pathway.** (**a**) Model of the upper α-ring and the adjacent regions of Rpn5 and Rpn6 are shown coloured according to their B-factors. For better orientation, the HbYX motifs of Rpt3 and Rpt5 are depicted in their binding pockets (green and cyan) and the orange arch corresponds to the position of Rpt3's region with highest B-factors. A significant increase in B-factors can be seen on the outer parts of α2, α3 and α 4 which all are adjacent to the Oprozomib binding subunit β5. The regulatory subunits Rpn5 and Rpn6, which directly bind the α-ring, show the highest B-factors, presenting a potential communication path indicated by flexible parts. (**b**) The model of the ATPase ring and the adjacent parts of Rpn5 and 6 are shown coloured by B-factors. The perspective onto the segment is the same as in **a**. A similar distribution of B-Factors as in **a** can be seen. Subunits Rpt2, 6 and 3 show increased B-factors as they are adjacent to the more mobile α-subunits in **a**. Similarly, the regulatory subunits Rpn5 and 6 show very high B-factors. Right: to analyse the symmetry of the ATPase, three conserved amino acids have been chosen in all six. The Cα-atoms have been connected and the inner angles of the resulting hexagons enclosed by Rpt6, Rpt2 and Rpt1 have been calculated. Whereas the C-terminal region of the ATPase forms a perfectly regular hexagon, the symmetry is clearly broken in the N-terminal region near the centre of the ATPase. This deviation from perfect symmetry indicates the required motion for ATPase activity and is consistent with an increase in model B factors. (**c**) Focused classification on the Rpn9/10 (orange) region and subsequent refinement in RELION revealed different conformational states for the receptor regions. Only exemplary conformations are shown. Similar conformations can be found by focusing on Rpn5 (blue) only.

α-subunits 1, 2, 3 and 4, exhibit an asymmetric B-factor elevation. In addition, a gradual vertical deviation from perfect six-fold symmetry is evident in the ATPase (Fig. 3b), which is even visible in a different sugar conformation of the bound nucleotide in Rpt2

(Supplementary Fig. 7). Furthermore, the motion of Rpn5 is correlated in the next vertical tier with the local conformation variations of the ubiquitin receptor Rpn10 (Fig. 3c, Supplementary Movie 2). Focused 3D classification reveals

considerable mobility of Rpn10, which in some conformers is in direct contact to the coiled-coil region of Rpt4/5. In other conformers, Rpn10 completely detaches from the deubiquitinase Rpn8 (Supplementary Movie 3).

In conclusion, we have provided evidence that 20S proteasome inhibitors allosterically regulate the RP to adopt a non-rotated (presumably inactive) conformation. To our best knowledge, allosteric effectors that have such an extended reach are unprecedented and have not been described yet. We have presented evidence that the RP subunit Rpn5 most likely is the lever, which conveys the information about inhibition of the proteolytic active site over a long-range distance. This is supported by the notion that Rpn5 is an essential lid protein (so far described in yeast and plants)[11]. Previous studies at lower resolution have shown that the rotated (presumably active) state is preferentially adopted by RP binding to a slowly degraded substrate[4], or by the addition of slowly hydrolyzable nucleotide[6] to the yeast holoenzyme. Ligand binding to either the RP or the CP has distinctly opposing effects on the conformational motion of the RP, which suggests a feedback regulation between RP and CP through Rpn5. This knowledge enforces the notion that development of 20S inhibitors with novel binding sites and inhibition chemistries will have a profound impact on the allosteric regulation of the proteasome holoenzyme. Vice versa, inhibitors that target the conformational variability of the RP will profoundly influence the catalytic activity of CP active sites. We postulate that allosteric regulation of large macromolecular complexes, by catalytic active site small molecule inhibitors is a general feature.

## Methods

**Materials.** Standard chemicals were obtained from Sigma Aldrich (Taufkirchen, Germany). Oprozomib and Epoxomicin were purchased from ApexBio (Houston, USA). The crosslinking agent BS3 was obtained from Thermo Scientific (Waltham, USA).

**Protein purification.** S30 HeLa cytoplasmic extract[12] was prepared by hypotonic lysis, centrifuged at 30,000g for 30 min at 4 °C, flash frozen in liquid nitrogen and stored at − 80 °C. The S30 extract was thawed in a water bath at 37 °C, supplemented with purification buffer to 1 × from a 10 × stock, sucrose powder to 20% (w/v), octyl glucose neopentyl glycol (from a 10% (w/v) stock solution in water) to 0.1% (w/v), iodacetamide to 10 mM, N-ethylmaleimide to 10 mM, benzamidine chloride to 10 mM and ATP to 7.5 mM. The extract was incubated at room temperature on a magnetic stirrer for 30 min, followed by an addition of Dithiothreitol (DTT) powder to 50 mM and a second incubation at room temperature for 30 min. The S100 extract was prepared by centrifugation at 100,000g for 2 h at 4 °C and the supernatant was filtered through three layers each of cheese cloth and miracloth.

The S100 extract was processed by two subsequent rounds of precipitation with PolyEthyleneGlycol400 (PEG400; number signifies the mean molecular weight of the PEG polymer). First, PEG400 was added to a concentration of 23% (v/v) to the S100 extract at 18 °C on a magnetic stirrer and incubated for 30 min. Second, the supernatant was precipitated by raising the concentration of PEG400 to 30% (v/v) as described before. The precipitate contains the human 26S/30S proteasomes and was resuspended with purification buffer supplemented with 7.5 mM ATP, 5 mM DTT and 0.01% (w/v) lauryl maltose neopentyl glycol (LMNG) in an orbital shaker at 18 °C. The resuspended material was incubated with an ATP regeneration system (10 mM sodium creatine phosphate, 5 μg ml − 1 creatine kinase) at 30 °C for 30 min.

The sample was loaded on 20%/50% two-step sucrose cushions in purification buffer containing 7.5 mM ATP and 5 mM DTT. The cushions were centrifuged at 260,000g for 14 h at 4 °C, harvested in 500 μl fractions with Äkta Prime (GE Healthcare, Munich, Germany) and analysed by SDS–polyacrylamide gel electrophoresis (PAGE) to identify fractions containing 26S and 30S proteasomes. Fractions were pooled and precipitated by the addition of 40% (v/v) PEG400 for 30 min and after centrifugation (30,000g, 30 min), the precipitate was resuspended in purification buffer containing 5% sucrose, 7.5 mM ATP, 5 mM DTT and 0.01% (w/v) LMNG. The proteasomes were treated with Oprozomib at a concentration of 0.5 mM at 25 °C for 30 min. Proteasomes were loaded on linear 10–40% (w/v) sucrose gradients in purification buffer containing 7.5 mM ATP, 5 mM DTT, which were centrifuged at 220,000g for 16 h at 4 °C. In total, 400 μl fractions were analysed by SDS–PAGE, selected proteasome fractions were precipitated by the addition of 40% (v/v) PEG400 and resuspended in purification buffer containing 7.5% (w/v) sucrose, 7.5 mM ATP, 5 mM DTT and 0.01% (w/v) LMNG. As a final

step, proteasomes were fractionated on linear 10–45% (w/v) sucrose gradients in purification buffer containing 7.5 mM ATP, 5 mM DTT, which were centrifuged at 260,000g for 16 h at 4 °C. Fractions containing 26/30S proteasomes were yet again identified by SDS–PAGE, precipitated by the addition of 40% (v/v) PEG400 and resuspended in 2 × purification buffer containing 15% (w/v) sucrose, 15 mM ATP, 10 mM DTT and 0.02% (w/v) LMNG yielding the final purified protein preparation at 30 mg ml − 1. Protein concentrations were determined by the Bradford assay (BioRad, Munich, Germany) using BSA as a standard. This purification procedure reproducibly yields 45 mg purified human 26/30S proteasomes, starting from 800 ml S100 HeLa cytoplasmic extract at a concentration of 10 mg ml − 1.

Purification buffer: 0.05 M Bis-Tris pH 6.5, 0.05 M KCl, 0.01 M MgCl₂, 0.01 M β-Glycerophosphate

**Negative staining EM sample preparation and image analysis.** Proteasomes were either supplemented with 2 mM Epoxomicin, 2 mM Oprozomib or DMSO as a control. After 30 min of incubation on ice, the respective samples were loaded on sucrose gradients (10–30% w/v sucrose). Gradient centrifugations were carried out in a TH660 Rotor (Thermo Scientific, Osterode, Germany) at a centrifugational force of 114,000g for 16 h at 4 °C. In total, 200 μl gradient fractions were collected.

EM grids of 26S proteasome fractions were prepared by floating a continuous carbon film in the solution for 1 min at 4 °C and staining with a saturated uranyl formate solution. Samples were imaged on a Philips CM200 microscope at a magnification of × 88,000 corresponding to a pixel size of 2.5 Å per pixel. In total, 500 micrographs per sample were collected in spot scanning mode using a TVIPS CCD camera.

Particles were selected and CTF correction was performed on the individual particle level. Resulting particles were subjected to several rounds of two-dimensional classification to remove images without particles and images containing contaminations such as ice crystals. The remaining particles were aligned against a 3D model of the rotated and independently to a model of the non-rotated state and assigned to the better fitting model according to the cross correlation. This was repeated in three iterations. After each iteration, new volumes were reconstructed from the assigned particles, low-pass filtered to the same resolution and normalized. Particles contributing to the individual classes were counted.

**Cryo-EM sample preparation.** BS3 (2.5 mM) was added to the purified proteasome holoenzyme (12 mg ml − 1) and incubated at 4 °C for 2 h. The crosslink reaction was terminated by the addition of 10 mM sodium aspartate (pH 6.5) and loaded on a GraFix gradient[13] (10–30% w/v sucrose, 0–0.1% glutaraldehyde). The gradient centrifugation was carried out in a TH660 Rotor (Thermo Scientific, Osterode, Germany) at a speed of centrifugational force of 114,000g for 16 h at 4 °C. A total of 200 μl gradient fractions were collected and immediately quenched by adding 20 mM of sodium aspartate (pH 6.5). The protein peak in the gradient fractions was assessed by a dotblot with Amidoblack staining. Peak fractions were analysed by negative staining EM. The fractions containing the single capped 26S proteasomes were buffer exchanged to the purification buffer without any sucrose using a Zeba spin column (Thermo Scientific, Osterode, Germany). The particles were adsorbed to a continuous carbon film for 1 min at 4 °C, attached to a Quantifoil (3,5/1) (Quantifoil, Jena, Germany) grid and freeze plunged in a Leica EM GP (Leica, Wetzlar, Germany) employing the blotting sensor at 75% humidity and 4 °C.

**Cryo-EM data acquisition.** The grids were imaged in a Titan Krios (FEI, Eindhoven, The Netherlands) (Supplementary Data 7) equipped with a Cs-Corrector (CEOS, Heidelberg, Germany) on a Falcon II detector. Images were taken at a nominal magnification of × 110,000, corresponding to a pixel size of 1.27 Å per pixel. The total dose (50 electrons per Å²) was fractionated on 17 frames. The first frame revealing inhomogeneous illumination due to the camera shutter was discarded. In total, 18,707 micrographs were collected in total (Supplementary Fig. 8).

**Image processing.** Individual image frames were aligned and weighted according to electron dose using the software unblur[14] to reduce the effects of drift and charging. The CTF of the remaining micrographs was determined using Gctf[15]. Particles were selected in a template-free manner, using image statistical properties in combination with mass centring. Individual particle coordinates were additionally refined by alignment against twelve low resolution reference images representing different views of the proteasome (Supplementary Fig. 9).

Subsequently, we performed several image sorting steps to remove contaminations, blurred images and bad particles. In a first step, power spectra were calculated, for each particle and classified using a hierarchical clustering scheme. The resulting class averages were visually inspected for Thon ring appearance and particles belonging to strongly charged or blurred classes were discarded. Second, several rounds of multi-reference alignment and two-dimensional classification were performed. Particles belonging to classes that did not show clear molecule views were discarded.

After having applied these image sorting procedures, the best class averages were used to generate an initial 3D model using simple PRIME[16]. This 3D model

was used as an initial reference in a 3D classification in RELION[17], which we used to classify the particles according to their two main conformational states. To ensure correct class assignment, all particles were aligned competitively against averaged maps obtained for the two main states. The flexible protein Rpn1 interferes with the alignments and therefore its density was masked out.

Particle images belonging to the non-rotated state of the proteasome were refined by RELION auto-refine. Subsequent hierarchical sorting discriminated further sub classes of various RP conformations. Specifically, a series of 3D classification steps without alignment using increasingly smaller masks was performed in RELION. In the first classification step, we used a mask for the whole proteasome holoenzyme excluding Rpn1, in the second iteration we used a mask for the whole RP (19S) subcomplex, in the third iteration a mask for the whole lid and finally a mask for Rpn2 only. The remaining particles (233,000) were refined to a final resolution of 3.8 Å and B-factor corrected in RELION. To further improve the map, particle polishing[18] was performed on the final particle stack in RELION.

A local resolution map was calculated in ResMap[19] by calculating local FSC values in a sphere with a diameter of 13 voxels moving over the entire 3D volume. In addition, the signal of CP was subtracted from the raw particles[20]. These subtracted particles were centred and again refined in RELION. Masks for the Rpt2/6, Rpn9/10 and Rpn5/6 regions were created in Chimera and focussed 3D classification without alignment was performed on the computationally generated 19S particles. Resulting 3D classes were refined.

**Conformational landscape analysis of the RP subunit.** A comprehensive explanation of the method is given in the Supplementary Note 1 of this manuscript. In brief, more than 346 3D class averages were obtained in RELION revealing conformational differences in the RP. The conformational variability was analysed quantitatively using PCA Eigenvolumes. By determining the linear factors of each volume towards those Eigenvolumes an energy landscape was calculated in MATLAB. The MATLAB scripts can be provided upon request.

**Model building.** The initial atomic coordinate model for the 20S particle was taken from a crystal structure of the Oprozomib-inhibited human 20S complex (PDB 5LEY)[10]. Models of each RP (19S) subunit were generated with Robetta[21,22] and docked as rigid bodies into the EM density map with UCSF Chimera[23]. The six nucleotides of the ATPase subunits were placed by fitting the crystal structure of PAN (PDB 3H4M)[24] into our density. Additional aid for regions, which had to be modelled at least partly *de novo*, was obtained using the secondary structure prediction server psipred[25].

An initial rigid body refinement was performed using real space refinement in Phenix[26] and subsequent manual modelling in coot[27]. Next, secondary structure restraints were generated using *phenix.ksdssp*. All secondary structure restraints were visually inspected and additional restraints were added manually. Several iterative rounds of real space refinement in Phenix and manual modelling in coot followed, where the last Phenix refinements included ADP refinement to calculate B-factors.

The present map quality does not allow to distinguish clearly between ATP and ADP in the ATPase and hence we modelled all nucleotides as ADP. In addition, to account for local resolution differences in the EM density map, we used calculated model B-factor distributions as a guideline to define the level of structural details interpreted in the final model (Supplementary Fig. 5). Accordingly, we analysed B factors in segments of five amino acids. Side chains were only modelled if the mean atomic B-factor per segment was smaller than 110 $\text{Å}^2$, segments with mean B-factors between 110 and 150 $\text{Å}^2$ were truncated to poly-alanine. Residues with mean B-factors higher than 150 $\text{Å}^2$ were not included in the final deposited PDB model.

**Data availability.** EM density maps have been deposited in the EMDB with accession number 4,146. Raw micrographs have been uploaded to EMPIAR database. Modelled atomic coordinates have been deposited in the Protein Data Bank with the accession number 5M32. The data that support the findings of this study are available from the corresponding authors upon request.

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

## Acknowledgements

We thank Thomas Conrad for HeLa cell growth and Hossein Kohansal for HeLa cell extract preparation. Monika Raabe and Henning Urlaub for mass spectrometry service. Mario Lüttich, Boris Busche, Jan-Martin Kirves, Georg Bunzel and Lukas Schulte for development of image processing software. Wen-ti Liu for graphical design support and Sabrina Fiedler for help with MATLAB and Brenda A. Schulman and Lars Bock for fruitful discussions. This work was funded by grants of the Deutsche Forschungsgemeinschaft (DFG; CH1098-1/1 to A.C., SFB860-TP A5 to H.S.). H.S. and A.C. received support by an R&D Instruct grant as part of the European Strategy Forum on Research Infrastructures (ESFRI), which is supported by national member subscriptions.

## Author contributions

J.S. and A.C. developed and performed proteasome purification. D.H. performed electron microscopy, image processing. A.C. and F.H. screened several proteasome inhibitors. D.H. and F.L. performed the energy landscape analysis and model building. A.C. and H.S. designed and supervised research. All authors contributed to the preparation of the manuscript.

## Additional information

**Competing interests:** The authors declare no competing financial interests.

**Publisher's note**: 

