## [Peer Review File · Nature Communications]

Reviewers' Comments:

Reviewer #1 (Remarks to the Author):

The manuscript contributed by Haselbech, Schrader et al. explores the conformational effects of Oprozomib, a proteasome inhibitor, on the 26S proteasome. The question of whether protease inhibitors confer long-range rearrangements to the 20S pore or beyond has long been debated in the proteasome field, and the authors contribute a compelling model for such allostery. The authors solved cryoEM structures of the 26S proteasome with and without ligand bound, and performed an extensive 3D analysis on the regulatory particle to assess the effect of ligand on conformational heterogeneity. Surprisingly, Oprozomib-binding profoundly limits the motions of the regulatory particle, thereby preventing substrate degradation. The authors suggest that Rpn5, a subunit that directly contacts the 20S alpha ring and is known to be an important component during the transition of the regulatory particle from one state to the another, conformationally transmits inhibitor binding.

The authors are experts in cryoEM analysis, and the methodology used to analyze the heterogeneity of the regulatory particle is perfectly suited to assess this biological phenomenon. The manner in which these data are presented is novel and visually striking. The long-range communication between the B5 protease active site and the periphery of the regulatory particle, while previously hypothesized and disputed, has not yet been described structurally. This manuscript presents convincing evidence and a potential mechanism for ligand-induced silencing of the regulatory particle. This work will serve as the basis for future biochemical and biophysical studies in numerous labs, and I heartily support its publication in Nature Communications.

Prior to publication, I request that the authors address an important point that they fail to discuss in the current form of the manuscript. This relates to the previously observed stabilization of polyubiquitinated substrates upon ligand binding. In the non-rotated state, the active site of the deubiquitinase Rpn11 is blocked from interaction with isopeptide linkages. In this state, substrates will bind and subsequently detach from the 26S without cleavage of the ubiquitin moieties. The kinetics of ubiquitin binding are not likely affected by maintenance of the non-

rotated state, explaining the authors' observation that ubiquitin does not accumulate on the proteasome. Rather, the proteasome is unable to rotate to assume the conformation that makes ubiquitin cleavage possible.

A minor issue I have with the manuscript is that extended data figure 5 is impossible to digest, and it is unclear how these data are informative to a reader, as there doesn't seem to be any particular organization or ordering associated with the listed structures. Perhaps if the densities were clustered, or representative densities were overlaid on the energy landscape this figure would be of interest to the reader.

Reviewer #2 (Remarks to the Author):

Haselbach et al. present an analysis of the structural and conformational differences in the 26S proteasome that occur upon binding of the 20S proteasome inhibitor Oprozomib. They determine high-resolution structures by cryo-EM with and without Oprozomib bound and present the conformational energy landscape for these two complexes by performing 3D classification of the different conformations produced.

I commend the authors for the creative nature of this study. I believe it should be published in Nature Communications after making the following changes:

1) Data (not a cartoon) should be shown to support the statement on lines 44-47 that the non-rotated state is stabilized by Oprozomib.

2) The figure that compares structures with and without Oprozomib to show the binding of the drug appears to compare maps at different resolutions, which can alter the appearance of the maps. Maps at the same resolution should be compared to show that Oprozomib binding leads to the appearance of the density identified as the drug.

3)Most importantly, the analysis that leads from quantification of the number of particle images in different 3D classes corresponding to different conformations to the energy landscape for the complex is not explained in sufficient detail. The methods section simply says the energy landscape was calculated in MATLAB from the linear factors of the Eigenvolumes. Information on how this analysis is performed and a discussion of potential shortcomings should be included. For example, do populations of 3D classes necessarily correspond to populations of molecules in the corresponding conformation? What can lead to the breakdown of this assumption?

4)Line 131: I believe the phrase “allosteric effectors, which have such an extended reach, are unprecedented...” should be “allosteric effectors that have such an extended reach are unprecedented...”.

Please find a detailed point to point discussion of all issues raised by the reviewers below:

Reviewer #1

Prior to publication, I request that the authors address an important point that they fail to discuss in the current form of the manuscript. This relates to the previously observed stabilization of polyubiquitinated substrates upon ligand binding. In the non-rotated state, the active site of the deubiquitinase Rpn11 is blocked from interaction with isopeptide linkages. In this state, substrates will bind and subsequently detach from the 26S without cleavage of the ubiquitin moieties. The kinetics of ubiquitin binding are not likely affected by maintenance of the non-rotated state, explaining the authors' observation that ubiquitin does not accumulate on the proteasome. Rather, the proteasome is unable to rotate to assume the conformation that makes ubiquitin cleavage possible.

The reviewer raises a highly interesting question about the correlation of polyubiquitinated substrate binding and the adoption of rotated and non-rotated conformational states of the regulatory particle. We had already explicitly addressed this point by an experiment depicted in Supplementary Figure 1 of our manuscript. To further address the reviewer's concerns, we have modified the Results section of our manuscript by the following sentences: "The treatment of the 26S proteasome with 20S proteasome inhibitors has been described to elicit a stabilization of polyubiquitinated substrates⁸, which may remain bound to the proteasome holoenzyme. We therefore investigated if polyubiquitinated substrates are accumulated in our proteasome preparations upon inhibition by drugs using anti-ubiquitin Western Blot analysis. As shown in supplementary figure 1, we have found no profound accumulation of polyubiquitinated substrates in our inhibited 26S proteasome preparation over non-treated controls. This indicates no correlation between the accumulation of polyubiquitinated substrates and proteasome inhibition. As a consequence, the allosteric regulation of regulatory particle rotation described in this manuscript is exclusively dependent on inhibitor binding."

In addition, we also have introduced some sentences in the introduction section to address this issue: "In this manuscript, we have determined high-resolution structures of inhibited and non-inhibited endogenously purified 26S proteasomes. Inhibition of the proteasome allows a structure of higher resolution to be determined. Importantly, by employing a newly established image analysis procedure, we can show that inhibitor binding causes a long-range allosteric regulation of the proteasome holoenzyme. Surprisingly, both species bind similar amounts of polyubiquitinated substrates as shown by Western Blot analysis. We can therefore conclude that the observed long-range conformational changes of the proteasome upon drug binding are exclusively due to the binding of the inhibitor Oprozomib."

Aside from this, the reviewer raises the issue that substrate binding and accommodation for deubiquitination and subsequent translocation into the proteolytic chamber are two separate issues. The reviewer suggests that substrate accommodation for deubiquitination might be the effector, which causes the rearrangement of the regulatory particle into the rotated state. Based on the results presented in this manuscript, we would agree with the referee. However, as we have not performed any experiments in this regard and other scenarios are not to be neglected a priori, we prefer not to comment on this in the present manuscript. The experiments required to address this point would entail the preparation of homogenous complexes of 26S proteasomes with accommodated substrates and structure determination thereof. These experiments are foreseeable to require a huge effort and far exceed the scope of this manuscript.

[...] extended data figure 5 is impossible to digest, and it is unclear how these data are informative to a reader, as there doesn't seem to be any particular organization or ordering associated with the listed structures.

The data shown in extended data figure 5 is the result of a standard 3D classification, which is indeed difficult to comprehend on its own. Moreover, while the ensemble of 3D conformers present in an entire dataset might contain important functional information, it is normally not considered in current cryo-EM studies. Our study exactly aims towards this problem and presents a solution to harness this information, which is entirely novel and illustrates the data in a rational manner. The PCA based approach allows us to order these many different 3D structures, which in the end can be interpreted as an energy landscape. We provide all those individual 3D structures in the extended data figure to allow the reader to judge the quality of the input data and ultimately the 3D volumes depicted in extended data figure 5 represent the raw data for the PCA analysis. Because the PCA based sorting of a pool of variable 3D structures is novel, we would prefer to leave this figure in.

Reviewer #2

1) Data (not a cartoon) should be shown to support the statement on lines 44-47 that the non-rotated state is stabilized by Oprozomib.

We agree with the reviewer that the mentioned statement needs to be supported by real data. We apologize for the misunderstanding we have caused. Figure 1 g is a schematic representation of the major rotational modes of the RP and is intended to be a visual aid for the reader. The raw data, which indeed indicates that the non-rotated state is stabilized by Oprozomib (and Epoxomicin) is depicted in Figure 1h as a bar diagrams. We have also clearly indicated this in the figure legends to figures 1 g and 1 h to avoid any further confusion. We hope the reviewer will now find this suitable.

2) The figure that compares structures with and without Oprozomib to show the binding of the drug appears to compare maps at different resolutions, which can alter the appearance of the maps. Maps at the same resolution should be compared to show that Oprozomib binding leads to the appearance of the density identified as the drug.

The reviewer is right with his comment. We have therefore added an additional Supplementary Figure (Supplementary Fig. 9), in which the map was filtered to 4.8 Å. Additional density, which corresponds to the inhibitor is also clearly visible at that resolution. However, we would prefer to keep figure 1e as it is, since it reflects the best possible resolution, which can be attained both with and without Oprozomib.

3) Most importantly, the analysis that leads from quantification of the number of particle images in different 3D classes corresponding to different conformations to the energy landscape for the complex is not explained in sufficient detail. The methods section simply says the energy landscape was calculated in MATLAB from the linear factors of the Eigenvolumes. Information on how this analysis is performed and a discussion of potential shortcomings should be included.

Due to space restrictions we decided to limit the paragraph in the methods section to a general description. Methodological details are, however, provided in the supplement, where the method is discussed in much greater detail.

Following points are discussed in more detail in the Supplement:

- treatment of the datasets with and without the inhibitor
- fundamental definitions on principal component analysis
- description of the Eigenvolumes used for the analysis
- explanation of the interpolation of the landscapes
- discussion of the methodological errors and limitations
- the input data

For example, do populations of 3D classes necessarily correspond to populations of molecules in the corresponding conformation? What can lead to the breakdown of this assumption?

The reviewer raises a general problem in cryo EM. The particle images generally have a low signal to noise ratio. Thus, the assignment of an individual particle into a certain class will not be entirely error-free. This problem can be minimized by the use of maximum likelihood approaches in cryo EM, as we did in our analysis. The use of a maximum likelihood approach accounts for these assignment errors and can estimate the size of the population rather accurately. To further minimize the risk of sorting particles into wrong conformational classes, we also grouped conformationally similar 3D volumes. Of course, the assumption and confidence by which 3D classes correlate with conformational populations of the complex under study will increase with particle numbers and statistics in cryo EM analysis.

4)Line 131: I believe the phrase “allosteric effectors, which have such an extended reach, are unprecedented...” should be “allosteric effectors that have such an extended reach are unprecedented...”.

Done

Reviewers' Comments:

Reviewer #1 (Remarks to the Author):

I am very pleased with the revisions and apologize for not noticing that the authors had already addressed my concern in the original submission of the manuscript. I heartily recommend this manuscript for publication & congratulate the authors on these exciting results.

-gabe lander

Reviewer #2 (Remarks to the Author):

The authors have addressed my comments. I believe the manuscript is now ready for publication.